# A Multi-Framework of Google Earth Engine and GEV for Spatial Analysis of Extremes in Non-Stationary Condition in Southeast Queensland, Australia

**Hadis Pakdel** [1,*], **Dev Raj Paudyal** [2] , **Sreeni Chadalavada** [1], **Md Jahangir Alam** [1,3] **and Majid Vazifedoust** [4]

1  School of Engineering, The University of Southern Queensland, Springfield Lakes, QLD 4300, Australia; sreeni.chadalavada@usq.edu.au (S.C.); mdjahangir.alam@usq.edu.au (M.J.A.)
2  School of Surveying and Built Environment, The University of Southern Queensland, Springfield Lakes, QLD 4300, Australia; devraj.paudyal@usq.edu.au
3  Murray-Darling Basin Authority (MDBA), Canberra, ACT 2601, Australia
4  Water Engineering Department, University of Guilan, Rasht 4188958643, Iran; vazifedoust@guilan.ac.ir
*  Correspondence: hadis.pakdel@usq.edu.au

**Abstract:** The frequency and severity of extremes, including extreme precipitation events, extreme evapotranspiration and extreme water storage deficit events, are changing. Thus, the necessity for developing a framework that estimates non-stationary conditions is urgent. The aim of this paper is to develop a framework using the geeSEBAL platform, Generalised Extreme Value (GEV) models and spatiotemporal analysis techniques that incorporate the physical system in terms of cause and effect. Firstly, the geeSEBAL platform has enabled the estimation of actual evapotranspiration ($ET_a$) with an unprecedented level of spatial-temporal resolution. Following this, the Non-stationary Extreme Value Analysis (NEVA) approach employs the Bayesian method using a Differential Evolution Markov Chain technique to calculate the frequency and magnitude of extreme values across the parameter space. Station and global climate datasets have been used to analyse the spatial and temporal variation of rainfall, reference evapotranspiration ($ET_o$), $ET_a$ and water storage ($WS$) variables in the Lockyer Valley located in Southeast Queensland (SEQ), Australia. Frequency analysis of rainfall, $ET_a$, and water storage deficit for 14 stations were performed using a GEV distribution under stationary and non-stationary assumptions. Comparing the $ET_a$, $ET_o$ and ERA5 rainfall with station data showed reasonable agreement as follows: Pearson correlation of 0.59–0.75 for $ET_a$, RMSE of 45.23–58.56 mm for $ET_a$, Pearson correlation of 0.96–0.97 for $ET_o$, RMSE of 73.13–87.73 mm for $ET_o$ and Pearson correlation of 0.87–0.92 for rainfall and RMSE of 37.53–57.10 mm for rainfall. The lower and upper uncertainty bounds between stationary and non-stationary conditions for rainfall station data of Gatton varied from 550.98 mm (stationary) to 624.97 mm (non-stationary), and for ERA5 rainfall datasets, 441.30 mm (stationary) to 450.77 mm (non-stationary). The results demonstrate that global climate datasets underestimate the difference between stationary and non-stationary conditions by 9.47 mm compared to results of 73.99 mm derived from station data. Similarly, the results demonstrate less variation between stationary and non-stationary conditions in water storage, followed by a sharp variation in rainfall and moderate variation in evapotranspiration. The findings of this study indicate that neglecting the non-stationary condition in some hydrometeorological variables can lead to underestimating their amounts. This framework can be applied to any geographical area for estimating extreme conditions, providing valuable insights for infrastructure planning and design, risk assessment and disaster management.

**Keywords:** climate extremes; Google Earth Engine; geeSEBAL; non-stationary; GEV distribution

## 1. Introduction

Stationarity is an assumption that was historically made to simplify the already complex statistics required. The terms of the return period and return level give vital information for designing, decision-making and estimating the implications of climatic occurrences

under the assumption of a stationary climate. This was generally a safe assumption as our data records were shorter in the past; therefore, not much change occurred over that short period of time. Furthermore, researchers of the past did not have the same computational power that we now have, so they relied on simplifying assumptions to make the math manageable. In particular, the availability of longer data records and the changing climate introduce risks in persistently assuming stationarity. For a long time, infrastructure design approaches have depended on stationary return levels which presume that the frequency of extremes does not fluctuate over time [1]. The frequency of extremes, on the other hand, has been shifting and is expected to continue to change in the future [2], and studies have shown that hydrological records in some regions demonstrate non-stationarity in the form of growing or decreasing patterns as well as their combination. Hydrological parameters are considered stationary over time, but the stationary condition may no longer be applicable as climatic and human effects create non-stationary behaviour [3].

The difficulty of incorporating spatial information within extreme value analysis approaches has been one focus of current studies on framework development given that many extreme occurrences involve spatial processes [4,5]. As a result, non-stationary climatic and hydrologic extremes require models that can account for them [6,7]. Southeast Queensland (SEQ) is one of Australia's most flood-prone zones [8] and the Intergovernmental Panel on Climate Change (IPCC) has recognised it as one of the "hotspots" for climate change [2]. According to Mpelasoka et al. [9], the soil-moisture-based drought frequency in SEQ catchments is expected to rise by 80 percent by 2070. In a major catchment in eastern Australia, Zhang, Wang [10] projected that during the next 80 years, real ET will rise significantly while water storage will decline.

This study aims to investigate non-stationary scenarios of rainfall (Rain), actual evapotranspiration ($ET_a$) and water storage (WS) variables obtained through the global climate datasets and ground-based measurements across various stations in the Lockyer Valley in SEQ. To achieve this goal, a geeSEBAL algorithm and NEVA model were employed in both stationary and non-stationary conditions. GeeSEBAL was used to automatically estimate ET, allowing for validation against ET station data. Considering that geeSEBAL does not depend on any ground-level measurements as input data, it is expected that this tool will be beneficial for the examination of water balances for worldwide application, as well as for water resource management in areas with limited data [11]. They also mentioned that the purpose of geeSEBAL is to enhance the comprehension of the effects of land cover alterations on ET during the past few decades. The recent version of geeSEBAL utilises Landsat imagery and reanalysis data to calculate the time series of $ET_a$, demonstrating encouraging outcomes for regional-scale investigations conducted in areas with limited data availability [11].

Changes in water storage are expected to result in changes in precipitation and evapotranspiration patterns, subsequently affecting plant phenology [12,13]. Since the intensity of extremes including extreme precipitation events, extreme evapotranspiration events and extreme deficit of water storage depend on several factors, such as vegetation cover [14], relative humidity, evapotranspiration [15] and topography, identifying the drivers of extremes helps us quantify, predict and project extremes [16]. Given the complexity of heat and energy exchanges between the land and the atmosphere, quantifying ET is regarded as a difficult undertaking [17]. Our capacity to analyse the ET process has substantially increased as a result of the advancement of various remote sensing technologies, particularly the creation and use of multiple surface energy balance (SEB) models [18]. Although, ET cannot be measured directly from satellites [19], it can be calculated using Mapping Evapotranspiration at High Resolution with Internalised Calibration (METRIC) [20] and the Surface Energy Balance Algorithm for Land (SEBAL) [17,21]. The core mechanism of the SEBAL, which is based on choosing endmembers that reflect the extremes of the hot (dry) and cold (wet) pixels, estimates the near-surface temperature gradient (dT) [17,20]. Previously, the process of identifying hot and cold pixels was performed manually, but advancements in technology have made it possible to automate this task. One such technique, known as Calibra-

tion using Inverse Modelling at Extreme Conditions (CIMEC) [22], utilises the Normalised Difference Vegetation Index (NDVI) and Ts percentiles to automatically determine endmembers. In this study, a platform called geeSEBAL (https://github.com/et-brasil/geesebal (accessed on 15 June 2023)) was used. It combines the capabilities of Google Earth Engine (GEE) with the Surface Energy Balance Algorithm for Land (SEBAL) framework [11]. It is an application or tool developed to leverage the GEE platform's application programming interface (API) and serves as a powerful tool for conducting various analyses related to remote sensing and evapotranspiration estimation.

A Bayesian inference framework that supports both non-stationary and stationary estimations was proposed by Cheng et al. [23]. As numerous natural phenomena happen under a non-stationary context, the idea of non-stationarity has been shown to be quite helpful in the field of hydroclimatology for analysing extremes. Their study findings suggested that NEVA calculates extreme return levels and variables effectively. The persistence of hydroclimatic extremes is destructive to the economy, natural ecosystems, agriculture, infrastructures and human health [24–28]. Identifying design extremes at different recurrence intervals and durations can be accomplished by performing a frequency analysis of extremes and looking at changes in the return period of the extremes using NEVA [1]. This model allows us to incorporate shifting extremes in intensity and frequency analysis [23].

Generalised Extreme Value (GEV) distributions and the Log Pearson Type 3 (LP3) are frequently utilised to conduct intensity analyses of hydroclimatic extremes. These statistical distributions are widely used for studying the occurrence and magnitude of extreme events in hydrology and climate research. Previous research used the GEV distribution to create Temperature Duration Frequency (TDF) curves [28,29]. Additionally, the Australian Rainfall and Runoff (ARR) guideline [30] has recommended the use of the GEV distribution for calculating design floods and rainfalls.

The study aimed to achieve the following specific objectives: (1) To evaluate the actual evapotranspiration derived from geeSEBAL by feeding two different climate datasets (ground-based observations and global climate products); (2) to compare the spatiotemporal distribution of the $P$ and $ET_o$ derived from a global climate dataset with the results of the same variables derived from ground-based observation and determine the model accuracy; (3) to evaluate and map the spatiotemporal distribution of water storage derived from a lump water balance for the last 32 years by considering two climate datasets and comparing the results incorporating physical drivers in terms of cause and effect; (4) to analyse the intensity and frequency of rainfall extreme events, evapotranspiration extremes and water storage deficit extremes under both stationary and non-stationary conditions using the GEV model for the estimation of different return levels.

In this study, we propose an integrated framework that combines geeSEBAL, NEVA-GEV and spatial distribution analysis to evaluate the return levels of extremes, including extreme rainfall events, extreme evapotranspiration events and extreme water storage deficit. We believe that the physical disturbance within a catchment alters the underlying process and results in temporal fluctuation in parameter values. As hydroclimatic variables can significantly vary over time, space, and climate zones, it is necessary to employ accurate data with higher spatial resolution for variability analysis [31]. So, we provided the geeSEBAL results by incorporating high quality datasets and considering the physical system drivers and their relationship. The outcomes of the study could be useful in understanding the spatial variation of rainfall, evapotranspiration and water storage by incorporating both stationary and non-stationary assumptions, thereby assisting decision-makers in making informed decisions for disaster preparedness, emergency response, health care services and the selection of appropriate materials for infrastructure development.

## 2. Materials and Methods

Figure 1 shows the main methods and data sets used in this study. The steps in this study are as follows: (a) deriving $ET_a$ from the geeSEBAL algorithm and $WS$ from a lump water balance, (b) accuracy assessment of $P$, $ET_o$ and $ET_a$ derived from station

and global climate datasets, (c) spatial analysis of extremes, including *P*, *ET*ₐ and *WS* and (d) investigating the extreme events analysis using the GEV model.

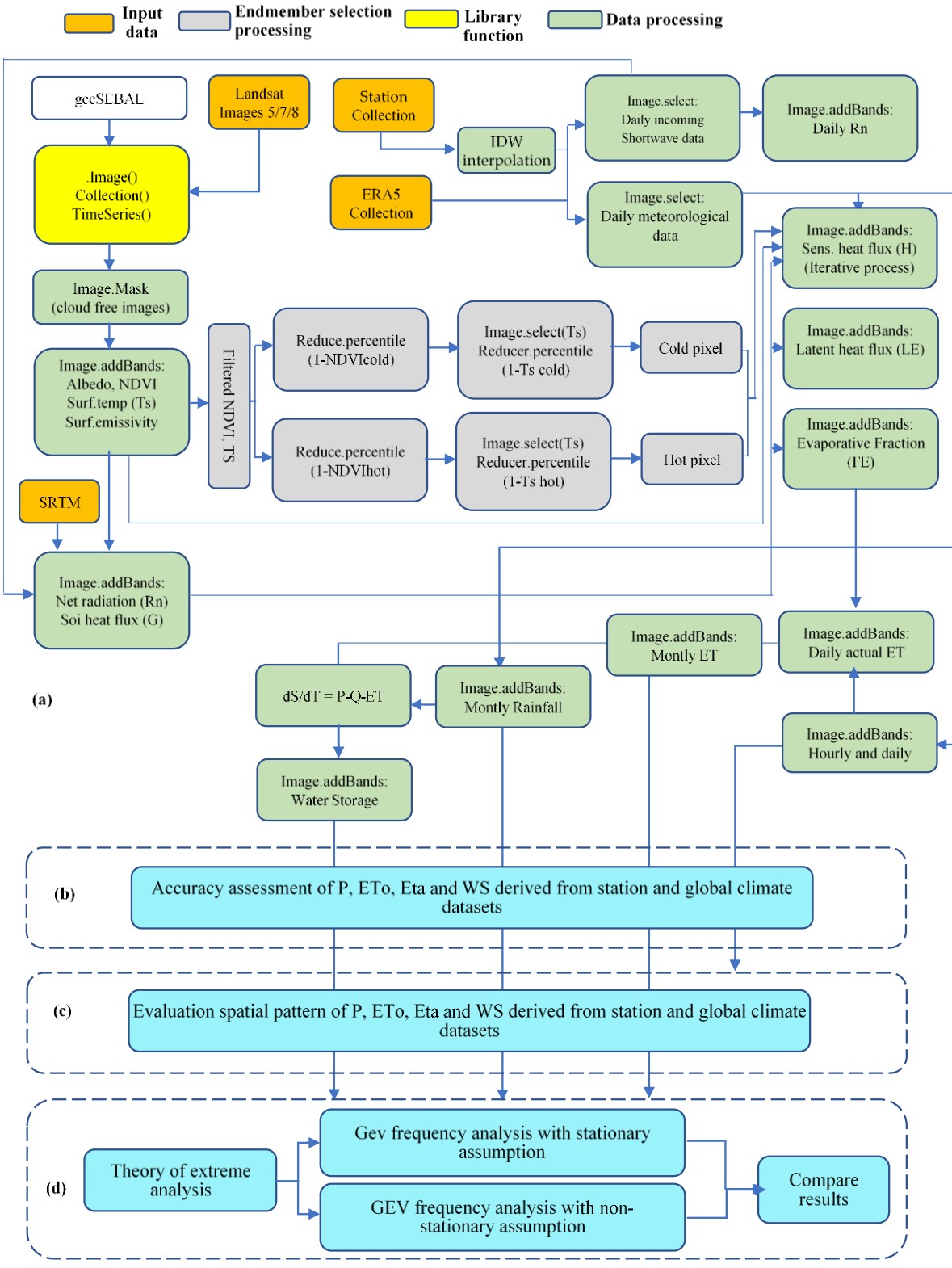

**Figure 1.** Flowchart of geeSEBAL, GEE and GEV model to estimate extremes. (**a**) geeSEBAL algorithm, (**b**) accuracy assessment of hydrometeorological variables from station and global climate data, (**c**) spatial pattern of hydrometeorological variables from station and global climate data, (**d**) extreme events analysis (Modified [11]).

*2.1. Study Region*

The Lockyer Catchment is the case study area which is located in SEQ as illustrated in Figure 2. Lockyer Creek is the main stream which runs eastward into the Brisbane River then enters Moreton Bay [32]. The catchment is located east of Toowoomba and west of Brisbane, within the local government boundaries of the Lockyer Valley Regional Council, Somerset Regional Council, Toowoomba Regional Council and Ipswich Regional Council [33]. Recognising its significance, the relevant infrastructure operators and decision-makers, such as the Queensland Department of Environment and Science and Seqwater, acknowledged the importance of this catchment [34]. It covers approximately 3000 km², with an average annual rainfall of 1000–2012 mm [35].

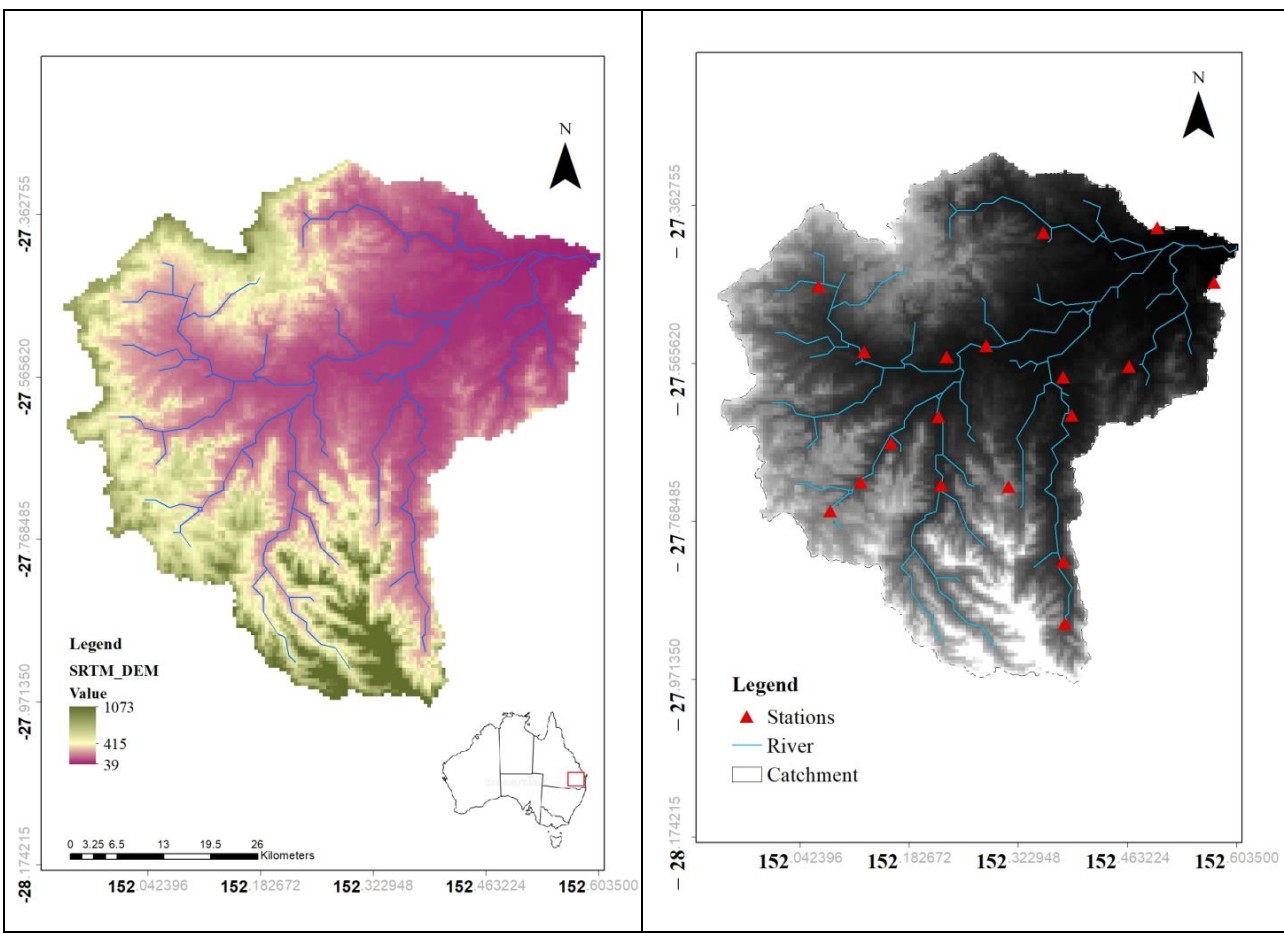

**Figure 2.** Geographical location of studied area in Australia (**Left**). Hydro-meteorological stations considered along the Lockyer Catchment (**Right**).

However, there is a significant temporal fluctuation, resulting in rivers that are dry for most of the year. The watershed comprises some of Australia's richest farmed areas, with high-value vegetable cultivation and grazing [32]. The northern and southern sections of the Lockyer watershed receive a lot of rain. The majority of the basin received moderate to low rainfall and during Australia's recent droughts and the Lockyer Creek valley was one of the driest catchments in Queensland [36]. Collectively, they drain around 3000 km² of land, about a fourth of the Brisbane River's watershed. The Lockyer Catchment, which has a population of over 35,000 people, has considerable environmental, economic and social importance.

In this region, there have been several unique climatic events in recent decades, such as between 1988 and 1989, when there was above average rainfall, and from 2000 to 2008, when there was a catastrophic and prolonged drought. Drought relief measures were

implemented in 2008. However, the region continues to experience reoccurring cycles of drought and flood crises, sometimes impacting the entire country for an extended duration. The frequency of flood events, like the one in 2022, may be misestimated under a stationarity assumption given that previous studies have shown the assumption to no longer be valid. Moreover, it has been observed that the relationship between runoff and rainfall in the Lockyer Catchment is non-stationary [37,38]. Therefore, it is crucial to develop novel methods to estimate non-stationary rainfall extremes, evapotranspiration extremes and water storage deficit to enhance the design and management of hydraulic structures which minimise human and financial losses in the future.

### 2.2. Observed and Global Climate Dataset

Daily meteorological data including minimum and maximum air temperature (°C), minimum and maximum relative humidity (%), wind speed (m/s), surface solar radiation (MJ/m$^2$) and hydrological data, such as evapotranspiration (mm) and rainfall (mm), were sourced from two datasets: ground-based observations (Figure 2) and global climate products. The ground-based observations were obtained from SILO, an Australian climate data source (http://www.longpaddock.qld.gov.au/silo (accessed on 15 January 2023)) [39,40] that covers the period from 1990 to 2022. SILO data are the most commonly used and most reliable climate data for environmental studies in Australia [41]. Fourteen SILO meteorological and hydrological stations were applied for this research. First, the Inverse Distance Weighting (IDW) interpolation method was used to interpolate the station data and rasterise the meteorological parameters. To run the geeSEBAL algorithm, daily meteorological ground-based observations and the hourly fifth generation ECMWF reanalysis (ERA5) climate dataset with 9 km spatial resolution were incorporated in GEE and were used separately for running geeSEBAL, as well as estimation of water storage. Land use information was obtained from the Australian government, Geoscience Australia (https://www.ga.gov.au/, accessed on 15 January 2023), for simulation purposes. A Shuttle Radar Topography Mission (SRTM) digital elevation model (DEM) [42] dataset with a grid size of 90 m was used as input for the geeSEBAL algorithm.

### 2.3. Google Earth Engine Application: The geeSEBAL Algorithm

The initial method created by Bastiaanssen et al. [17] is the foundation for the geeSEBAL [11], which makes the assumption that latent heat flux (*LE*) (W/m$^2$) can be approximated as a residual of the surface energy balance.

$$LE = R_n - G - H \tag{1}$$

where $R_n$ is the net radiation; $G$ is the soil heat flux (W/m$^2$); $H$ is the sensible heat flux (W/m$^2$). Given that both H and the aerodynamic resistance to turbulent heat transport ($r_{ah}$) are unknown, *H* was calculated by an iterative procedure.

$$H = \frac{\rho_a \, C_p dT}{r_{ah}} \tag{2}$$

where $C_p$ is the specific heat capacity, $r_{ah}$ aerodynamic resistance to turbulent heat transmission between two heights, dT represents the near-surface temperature difference between two heights $\rho_a$ is air density [17]. A linear connection between $T_s$ and (*dT*) is provided and cold and hot endmember selection is required to solve the iterative approach. The *a* and *b* coefficients are calculated individually for each picture.

$$dT = a + bT_s \tag{3}$$

Comprehensive documentation for the formulation and calibration of the SEBAL algorithm is covered in [17]. GEE infrastructure was used to develop the geeSEBAL algorithm [43], which enables the estimation of evapotranspiration at regional scales by using

meteorological reanalysis data and Landsat imagery [11]. The schematic representation of geeSEBAL is illustrated in Figure 1, highlighting the factors that were considered in the present study. The JavaScript APIs were used to integrate the SEBAL algorithm into GEE [11].

The three primary purposes of the geeSEBAL tool are: (1) Image: derivation of actual evapotranspiration from a particular image (accessible for JavaScript); (2) ImageCollection: batch method to calculate $ET_a$ provided a date range and (3) Timeseries: long-term $ET_a$ time series estimate at user-provided locations. All applications and codes are freely accessible at https://github.com/et-brasil/geesebal (accessed on 15 January 2023). Additionally, the Earth Engine programme (https://etbrasil.org/geesebal, accessed on 15 January 2023) offers a graphical user interface version of geeSEBAL [11,44,45]. For the purpose of running geeSEBAL, a series of Landsat images with the highest data quality were used. We used cloud cover filters using the CFMask method [46], which generates a bitmask to identify each image's pixels for clouds, clouds with shadows, clouds with confidence and pixels for ice and snow. The characteristics of Landsat collections used in this study are included in Table 1.

**Table 1.** The datasets available in the GEE platform that are used in geeSEBAL.

| Product | GEE ID | Resolution | Time Coverage |
|---|---|---|---|
| LANDSAT 8 OLI/TIRS | LANDSAT/LC08/C01/T1_SRLANDSAT/LC08/C01/T1 | 30 m | Apr/2013–Present |
| LANDSAT 7 ETM+ | LANDSAT/LE07/C01/T1_SRLANDSAT/LE07/C01/T1 | 30 m | Jan/1999–Present |
| LANDSAT 5 TM | LANDSAT/LT05/C01/T1_SR LANDSAT/LT05/C01/T1 | 30 m | Mar/1984–May/2012 |

The geeSEBAL algorithm was run separately using ERA5 reanalysis data and station data. Advised hot and cold endmembers percentiles were applied to determine the hot and cold pixels. The selection of candidates for the cold endmembers involved identifying densely vegetated areas, whereas sparsely vegetated regions were considered for the hot endmembers. The candidate with the lowest $T_s$ (20%) and greatest NDVI (5%) percentiles was selected to represent the cold end. Conversely, the hot endmember was determined by selecting the candidate with the highest Ts percentile (20%) and the lowest NDVI (10%) percentiles.

Long-term $ET_a$ was generated using the geeSEBAL algorithm, and the gaps between the Landsat images were filled by multiplying a ratio map of the last available simultaneous maps of $ET_a$ and reference evapotranspiration, $ET_o$, to the new $ET_o$ map [47]. It is vital to consider the non-stationary condition of rainfall and storage (Equation (4)) in a catchment, especially when the relationship between rainfall and runoff is non-stationary. In this equation, $P$ is daily precipitation (mm), $q$ is runoff (mm) 20% of precipitation [30,48], and $ET_a$ is daily actual evapotranspiration (mm) derived from the geeSEBAL algorithm.

$$\frac{dS}{dT} = P - q - ET_a \tag{4}$$

### 2.4. Assessing Extremes in a Non-Stationary Approach Using GEV Model

A stationary time series is defined as one in which all finite dimensional distributions are time invariant [49]. The stationary assumption may not be valid in relation to changes caused by human and climatic factors, which results in non-stationary situations [3]. The simplifying assumption of stationarity is used to estimate the largest instantaneous extremes, and structures are built with this assumption in mind. It is important to acknowledge that the impacts of climate change are growing, leading to an increase in non-stationary conditions worldwide. The frequency of flood events like the one in 2022 may be misestimated under a stationarity assumption given that previous studies [38] have

shown the assumption to no longer be valid. Therefore, it is critical to take a non-stationary approach to these issues (Figure 1). In order to analyse the non-stationary extremes, the NEVA software package [1] was utilised.

NEVA calculates the extreme value in a Bayesian method using a Differential Evolution Markov Chain technique for global optimisation across the parameter field [1]. Extreme Value Theory (EVT) provides two fundamental distributions for describing extremes: either the peaks-over-threshold approach using the Generalised Pareto Distribution (GPD) [50–52] or the block maxima (or minima) approach using the Generalised Extreme Value (GEV) family of distributions [53]. NEVA typically contains two sections: 1. GEV distribution for yearly maximum analysis (block maxima) and 2. The peak-over-threshold (POT) approach uses the GPD for analysis of extremes over a specific threshold. The GEV method has been applied in this study. The GEV distribution's cumulative distribution function (cdf) is represented using the following equations:

$$F\left(x\right) = exp\left\{-\left(1 + \varepsilon\left(\frac{x-\mu}{\sigma}\right)\right)^{\frac{-1}{\varepsilon}}\right\} \tag{5}$$

$$\left(1 + \varepsilon\left(\frac{x-\mu}{\sigma}\right)\right) > 0 \tag{6}$$

The GEV distribution is incredibly adaptable to simulate the behaviour of many severe occurrences since it just has three parameters $(\mu, \varepsilon, \sigma)$. The location, scale and shape parameters are defined by $\mu, \varepsilon$ and $\sigma$, respectively [52].

The GEV becomes the Gumbel, Fréchet and Weibull distributions, respectively, when $\varepsilon = 0$, $\varepsilon < 0$ and $\varepsilon > 0$, respectively.

In NEVA, the shape and scale parameters are assumed to be constant while the location parameter is considered to represent a linear function of time to accommodate for non-stationary [1].

$$\mu\left(t\right) = \mu_1 t + \mu_0 \tag{7}$$

NEVA uses the Mann–Kendall trend test at the user-selected significance level to identify trends and non-stationarity in extremes in data [1]. The Mann–Kendall (MK) statistical test [54] was undertaken to calculate the importance of climatic time series' trends [55–58]. There is no monotonic trend at the specified level of significance according to the null hypothesis (H0) of the MK test. The alternative hypothesis (Ha) in this test shows a monotonic trend over time. More details about Mann–Kendall are documented in [31]. The Bayesian approach employs a generic inference methodology. The Bayes theorem for estimating GEV parameters under the non-stationary assumption can be defined as follows under the assumption that observations are independent of one another [1,52]:

$$p\left(\beta\middle|\overrightarrow{y},x\right) \propto p\left(\overrightarrow{y}\middle|\beta, x\right)p((\beta|x)) \tag{8}$$

$$p\left(\overrightarrow{y}\middle|\beta, x\right) = \prod_{t=1}^{Nt} p(y_t|\beta, x(t)) = \prod_{t=1}^{Nt} p(y_t|\mu(t), \sigma, \varepsilon) \tag{9}$$

For the components, which are β without $x(t)$, the stationarity may be thought of as a specific instance of the aforementioned equation:

$$p\left(\theta\middle|\overrightarrow{y}\right) \propto p\left(\overrightarrow{y}\middle|\theta\right) p(\theta) = \prod_{t=1}^{Nt} p\left(y_t|\theta\right) p(\theta) \tag{10}$$

Thus, according to the non-stationary assumption, $x(t)$ stands for the collection of all covariate values. The derived posterior distributions $p\left(\theta\middle|\overrightarrow{y}\right)$ and $p\left(\overrightarrow{y}\middle|\beta, x\right)$ reveal the parameters' behaviour under stationarity $\theta = (\mu, \varepsilon, \sigma)$ or non-stationarity $\beta = (\mu_1, \mu_0, \varepsilon, \sigma)$.

## 3. Results

The research introduces a comprehensive approach by combining geeSEBAL with two climate data sets (global data and station data), NEVA-GEV and spatial distribution analysis. This integrated framework aims to assess the return levels of extreme events, such as heavy rainfall, evapotranspiration and water storage deficit, by taking into account the physical factors and their interactions within the system.

### 3.1. Station Data and ERA5 Land Reanalysis Feeding into geeSEBAL as Meteorological Inputs

GeeSEBAL was run by ERA5 reanalysis and station data with advised percentiles. The cold (wet) endmember was selected in the area with the highest concentration of vegetation, consisting of the coldest 20% $T_s$ of the top 5% vegetated pixels according to NDVI, and the hot (dry) endmember was chosen in the area with the lowest concentration of vegetation, consisting of the warmest 20% $T_s$ of the lowest 10% NDVI vegetated pixels. To identify the spatial pattern of four parameters among stations and ERA5 reanalysis are displayed in Figures 3 and 4 at Lockyer Catchment. These two figures provide better insight into how rain, $ET_o$, $ET_a$ and $WS$ are distributed over the catchment for the period of 2000 to 2022. Rainfall follows a similar trend in the upstream and downstream parts of the catchment during these years. The rainfall and $ET_a$ have almost identical trends which indicates the highest rise in the year 2010.

Between December 2010 and January 2011, flooding of historic proportions swept across large areas of Queensland due to prolonged rainfall [59], as illustrated in Figures 3 and 4. More than 78% of the state was declared a disaster zone, affecting 2.5 million people negatively and causing about 33 deaths [59]. The flood events affected approximately 29,000 homes and businesses, with an estimated damage cost of over $5 billion [59]. Interestingly, the estimation of spatial patterns derived from ERA5 land reanalysis are almost identical with station data for rain. Whereas in $ET_a$, ERA5 reanalysis overestimates the spatial variability compared to station data. Accordingly, water storage for the year 2010 confirms that the whole catchment reached its highest annually. The spatial pattern of figures for ERA5 reanalysis represents the more noticeable variation in the year 2005. The rain and water storage deficit in station data range from (~1200 mm) to (~1500 mm) and from (~250 mm) to (~500 mm), respectively (Figure 3). Similarly, Figure 4 shows the rain and water storage in ERA5, depicting the highest rise (~1500 mm) and (~500 mm) for years 2010 and 2020, respectively, while potential evapotranspiration has a remarkable increase from (~1450 mm) to (~1600 mm) in the year 2020 for both station and ERA5 reanalysis datasets. Water storage for both the station and ERA5 data remain the same, whereas $ET_a$ for the station and ERA5 exhibit variation from approximately 100 mm to 700 mm and from 400 mm to 1000 mm, respectively. ERA5 reanalysis overestimates $ET_a$ in comparison to the station.

So, the relationship between actual evapotranspiration and water storage is similar. Irrigated areas were concentrated alongside the streams and the Gatton area, which is known as the agricultural hub in Southeast Queensland. This also confirms the strong correlation between water storage and irrigated areas. The Lockyer Valley includes some of Australia's richest farmed area, with high-value vegetable cultivation, and, as can be observed, the highest water storage is concentrated in the irrigated areas [32]. So, the relationship between actual evapotranspiration and water storage is similar.

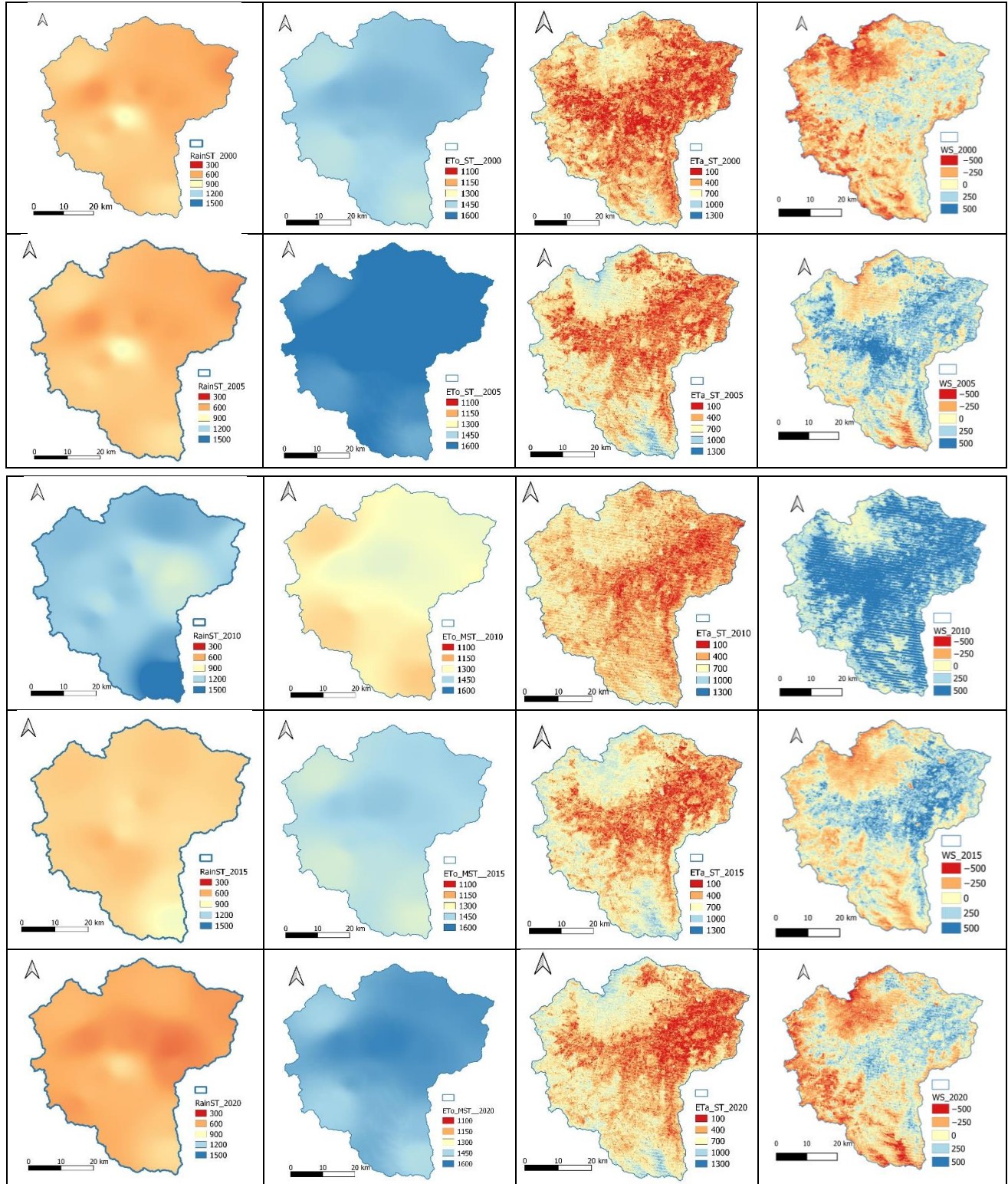

**Figure 3.** Changes in spatial pattern of hydroclimate parameters (annual cumulative rainfall (column 1), reference evapotranspiration (column 2), actual evapotranspiration (column 3) and water storage deficit (column 4)) from 2000 to 2020 based on geeSEBAL algorithm resulting from station datasets over the Lockyer Catchment.

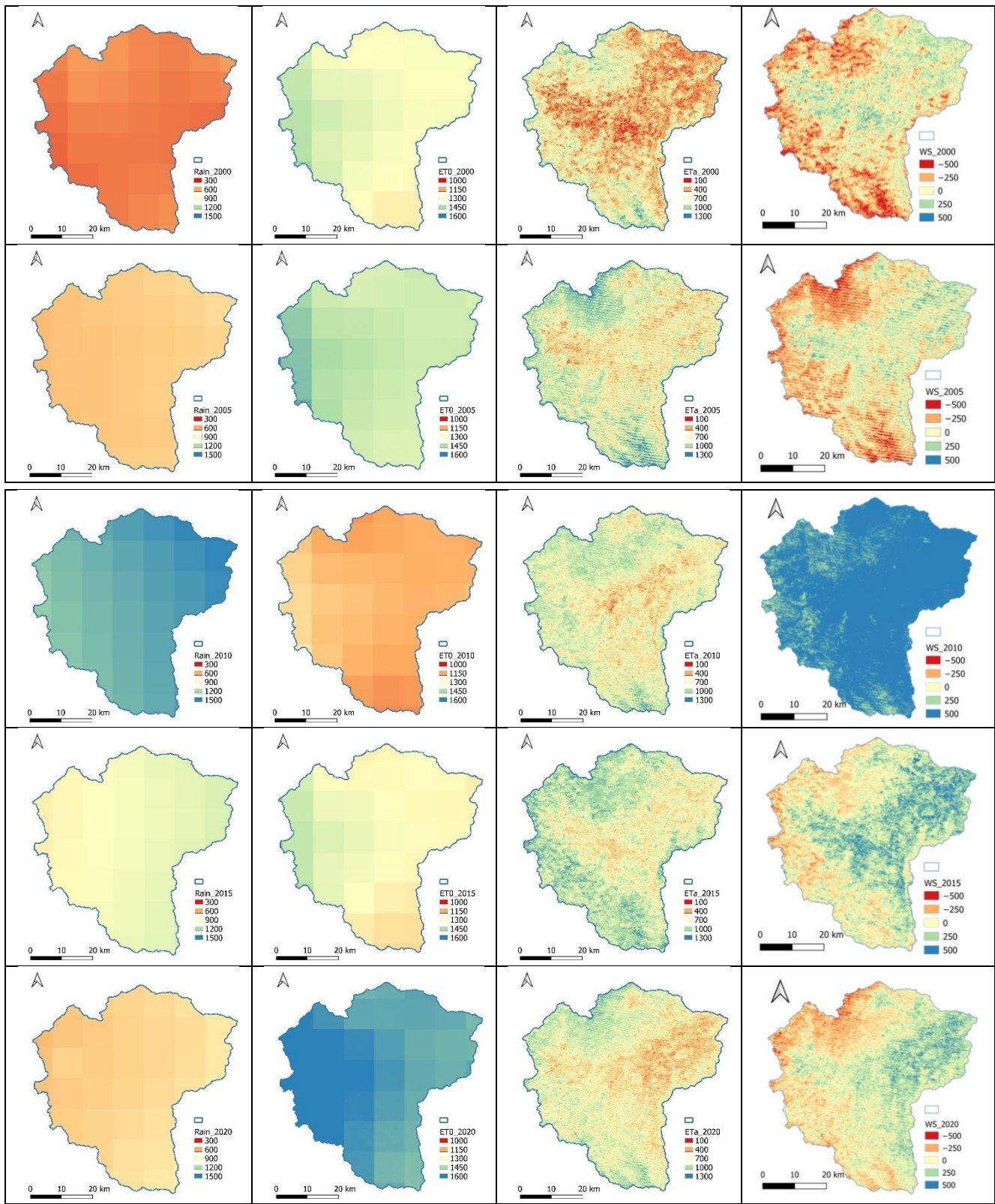

**Figure 4.** Changes in spatial pattern of hydroclimate parameters (annual cumulative rainfall (column 1), reference evapotranspiration (column 2), actual evapotranspiration (column 3) and water storage deficit (column 4)) from 2000 to 2020 based on geeSEBAL algorithm resulted from ERA5 reanalysis datasets over the Lockyer Catchment.

### 3.2. Validation of geeSEBAL $ET_a$ and $ET_o$ and Rain across Stations

To determine the model's accuracy, we compared the geeSEBAL algorithm driven by ERA5 reanalysis data and the reported data in the stations. Descriptive statistics results showed reasonable accuracy agreement between geeSEBAL $ET_a$, gridded $ET_o$ and ERA5 rainfall and stations (Table 2). All the stations presented $R^2$ and Pearson correlation for rainfall ranging from 0.76 to 0.84 and from 0.87 to 0.92, respectively, and some of the stations' results are presented in Table 2. The minimum RMSE was represented by Whitestone station (37.53 mm) and West Haldon station (38.97 mm) for rainfall. Due to the lack of clear sky images, $R^2$ values of $ET_a$ were low, ranging from 0.35 to 0.56 for the Upper Tenthill and Townson stations, respectively. Despite the low values of $R^2$, $ET_a$ presented a similar accuracy to rain and its RMSE, ranging from 45.23 mm (Whitestone) to 58.56 mm (Upper Tenthill). On the other hand, $ET_o$ estimates indicated a high accuracy for the 32 years, ranging from $R^2$ of 0.97 to 0.94, and Pearson's correlation of 0.97 to 0.96 for Townson and Placid Hills stations, respectively.

**Table 2.** Statistical comparison of annual cumulative rainfall, actual and potential evapotranspiration derived from geeSEBAL with the gauging station data.

| Station | Variable mm | $R^2$ % | Pearson's Correlation | RMSE mm/Month | Bias | MBias |
|---|---|---|---|---|---|---|
| Gatton | P | 0.76 | 0.87 | 45.38 | 1.14 | 0.16 |
| | $ET_a$ | 0.53 | 0.73 | 49.96 | −34.07 | 0.15 |
| | $ET_o$ | 0.94 | 0.96 | 83.73 | −71.33 | 0.28 |
| Placid Hills | P | 0.82 | 0.90 | 48.35 | −4.63 | 0.16 |
| | $ET_a$ | 0.42 | 0.65 | 54.34 | −38.45 | 0.13 |
| | $ET_o$ | 0.94 | 0.96 | 83.15 | −69.19 | 0.28 |
| Thornton | P | 0.84 | 0.92 | 44.09 | −9.04 | 0.17 |
| | $ET_a$ | 0.54 | 0.73 | 51.47 | −35.77 | 0.14 |
| | $ET_o$ | 0.94 | 0.97 | 80.25 | −64.27 | 0.27 |
| Townson | P | 0.84 | 0.92 | 57.10 | −19.70 | 0.17 |
| | $ET_a$ | 0.56 | 0.75 | 53.02 | −39.15 | 0.14 |
| | $ET_o$ | 0.95 | 0.97 | 77.29 | −62.01 | 0.26 |
| Upper Tenthill | P | 0.80 | 0.89 | 41.93 | −0.17 | 0.16 |
| | $ET_a$ | 0.35 | 0.59 | 58.56 | −45.15 | 0.12 |
| | $ET_o$ | 0.94 | 0.97 | 81.23 | −65.86 | 0.28 |
| West Haldon | P | 0.82 | 0.90 | 38.97 | −4.07 | 0.16 |
| | $ET_a$ | 0.51 | 0.72 | 49.14 | −33.76 | 0.15 |
| | $ET_o$ | 0.95 | 0.97 | 73.13 | −56.93 | 0.29 |
| Whitestone | P | 0.83 | 0.91 | 37.53 | 3.14 | 0.16 |
| | $ET_a$ | 0.54 | 0.74 | 45.23 | −30.51 | 0.15 |
| | $ET_o$ | 0.944 | 0.97 | 74.33 | −62.85 | 0.28 |

Figure 5a–u illustrates the nominated stations' Pearson correlation which revealed a strong spatial agreement with $ET_o$. For all 14 stations, $ET_o$ had remarkable agreement, ranging from 0.65 to 0.97. It also shows a similar relationship for the rain and $ET_a$, respectively.

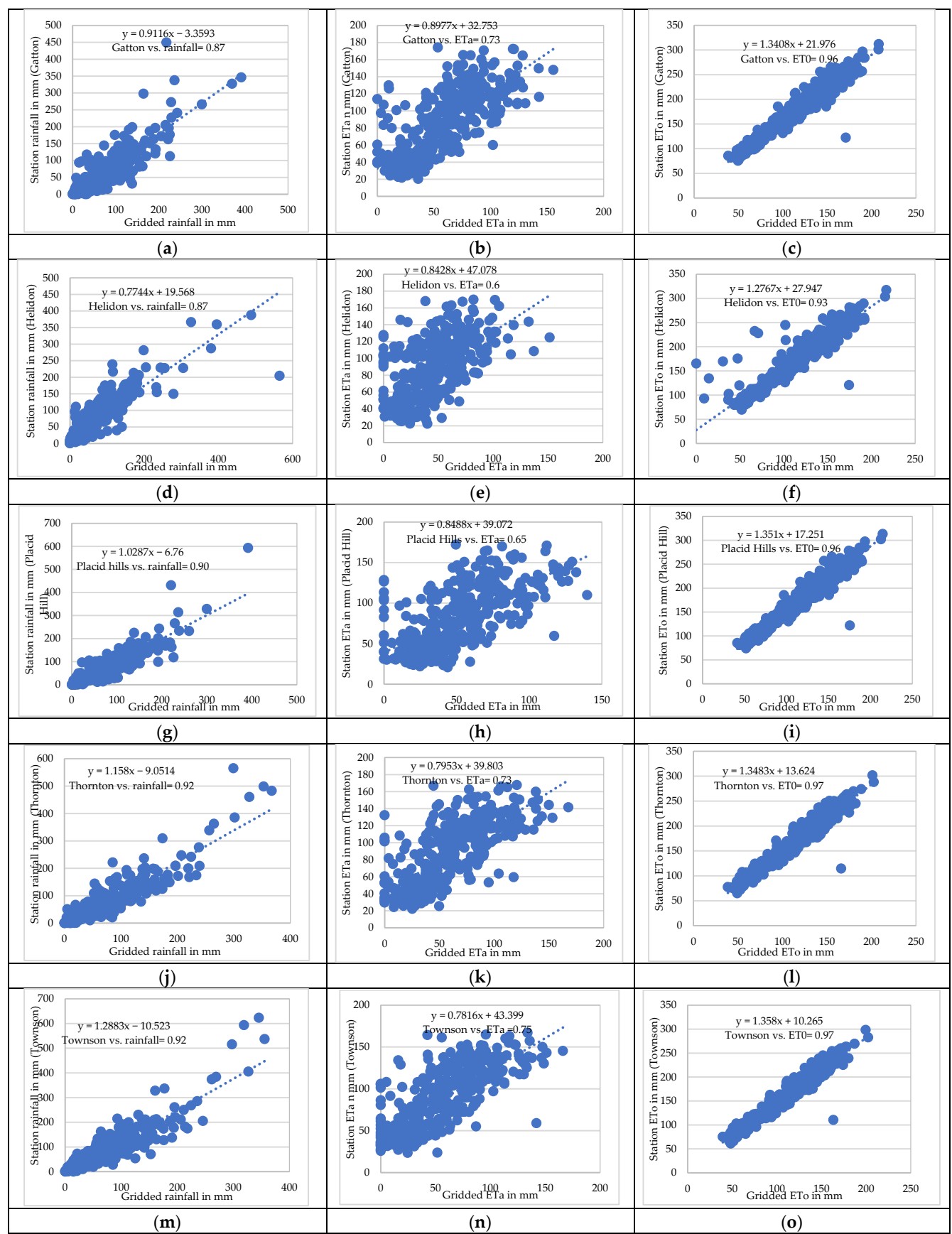

**Figure 5.** *Cont.*

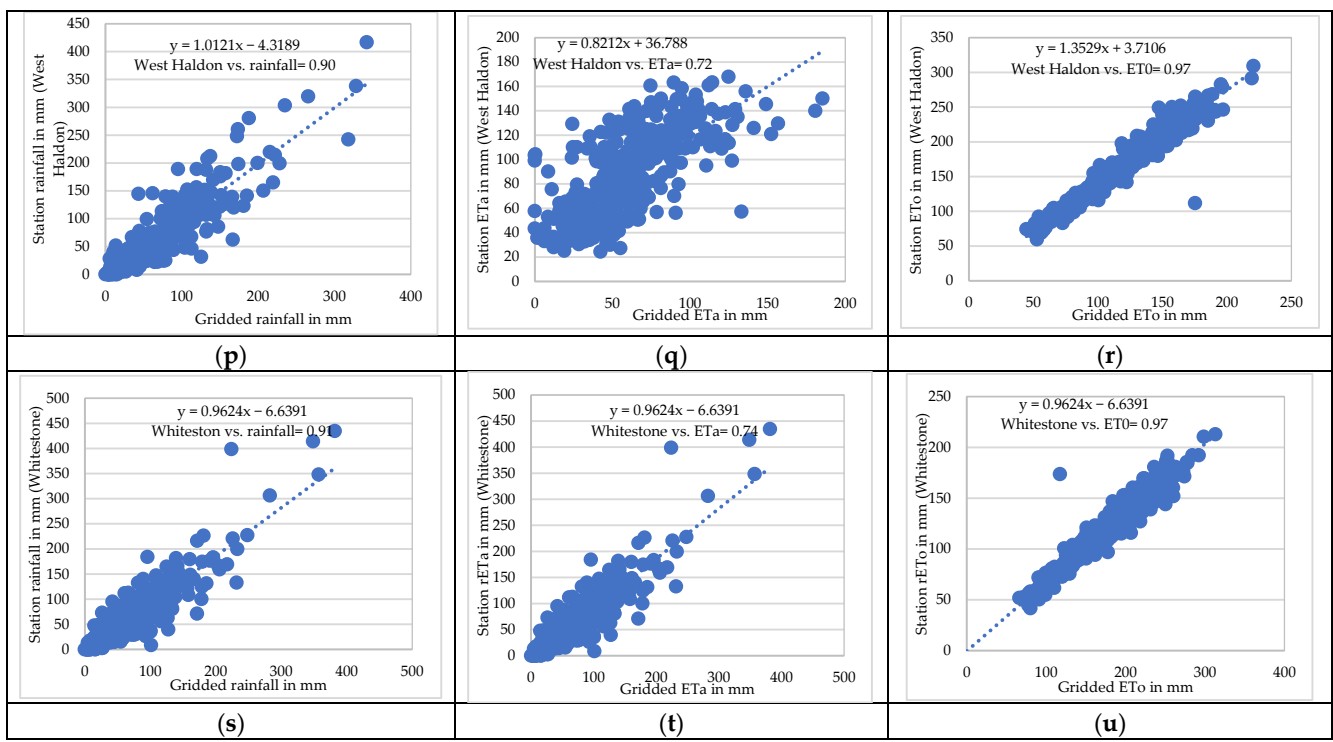

**Figure 5.** The relationship between monthly cumulative rainfall, reference evapotranspiration and actual evapotranspiration derived from geeSEBAL for the period 1990–2020 with the station datasets (**a**–**u**) in the Lockyer Catchment.

### 3.3. Results of Trends

The null hypothesis (H0) identified that there is no monotonic trend at the specified significance level (SL) in extremes. The alternative hypothesis (Ha) suggests that the data exhibit a trend at the SL. The SLs of 0.01 (1%), 0.05 (5%) and 0.1 (10%) were investigated in the calculation. The results in Table 3 show that there is a significant trend in the maximum precipitation series for the stations Placid Hills (5% SL) and Townson (1%) and the global climate dataset for Placid Hills (5% SL) and Townson (1% SL). Some stations and the global climate dataset exhibit a trend; therefore, they are in the non-stationary form.

**Table 3.** Detailed statistical results of trend and step change analysis of rain extremes for gauging stations in Lockyer Catchment.

| Test | Statistical Tests | *p*-Value | Test Statistic | Critical Values SL = 0.1 | SL = 0.05 | SL = 0.01 | Test Result |
|---|---|---|---|---|---|---|---|
| **Placid Hills** | | | | | | | |
| Trend detection | Mann–Kendall | 0.028 | −2.2 | 1.645 | 1.960 | 2.576 | H0 rejected at 5% |
| **Global climate dataset corresponding to Placid Hills** | | | | | | | |
| Trend detection | Mann–Kendall | 0.011 | −2.5 | 1.645 | 1.960 | 2.576 | H0 rejected at 5% |
| **Townson** | | | | | | | |
| Trend detection | Mann–Kendall | 0.009 | −2.62 | 1.645 | 1.960 | 2.576 | H0 rejected at 1% |
| **Global climate dataset corresponding to Townson** | | | | | | | |
| Trend detection | Mann–Kendall | 0.016 | −2.41 | 1.645 | 1.960 | 2.576 | H0 rejected at 5% |

### 3.4. Results of Stationary and Non-Stationary Analysis for Rainfall

Studying the non-stationary aspects of the extremes is the goal of this research. Table 4 shows the return levels of rainfall for the stations and global climate data for 14 selected stations in the Lockyer Catchment which are located in the southern part of Queensland. Return levels of extreme were estimated using the NEVA software for the return periods of

10 through 100 years, which are the standard design return periods used in hydrologic studies. The (100-year return period) and (10-year return period) station rainfall in Gatton and Helidon range from 550.98 mm to 312 mm (stationary), from 624.97 mm to 324.47 mm (non-stationary) and from 441.30 mm to 279.35 mm (stationary), from 450.77 mm to 282.12 mm (non-stationary). The return levels of Townson and Placid Hills which are located next to the Gatton agricultural hub of Southeast Queensland are indicated in Table 4. Placid Hills exhibits a variation of design rainfall for (20-year return period) to (50-year return period), from 371.52 mm (stationary) to 387.93 mm (non-stationary), and from 467.35 mm (stationary) to 491.09 mm (non-stationary), respectively. However, for gridded rainfall, Placid Hills indicates the highest variability of (50-year return period) from 354.20 mm (stationary) to 624.46 mm (non-stationary). Moreover, design rainfall of Placid Hills for (20-year return period) ranges from 305.19 (stationary) to 406.05 (non-stationary). According to Table 4, the extreme rainfall events under the non-stationary assumption are higher than the extreme rainfall events under the stationary assumption in both ground-based measurements and global climate data. The results also show that the difference between the maximum rainfall events under the stationary and non-stationary assumptions in the global climate data is generally greater than in the station data for different return periods.

**Table 4.** The maximum rainfall (mm) in different return periods for different stations in the Lockyer Catchment.

| | Rainfall | | Rainfall | |
|---|---|---|---|---|
| **Return Period** | **Station** | | **ERA5 Data** | |
| **Gatton** | **Stationary** | **Non-Stationary** | **Stationary** | **Non-Stationary** |
| 10 | 312.64 | 324.47 | 279.35 | 282.12 |
| 20 | 379.38 | 402.04 | 325.41 | 328.86 |
| 50 | 472.53 | 520.23 | 388.93 | 394.75 |
| 100 | 550.98 | 624.97 | 441.30 | 450.77 |
| **Helidon** | | | | |
| 10 | 339.96 | 336.66 | 292.61 | 325.507 |
| 20 | 437.19 | 436.11 | 343.12 | 393.58 |
| 50 | 596.99 | 606.25 | 417.67 | 492.67 |
| 100 | 748.07 | 765.74 | 478.62 | 575.84 |
| **Placid Hills** | | | | |
| 10 | 310.52 | 323.20 | 267.89 | 298.30 |
| 20 | 371.52 | 387.83 | 305.19 | 406.05 |
| 50 | 467.35 | 491.09 | 354.20 | 624.46 |
| 100 | 560.60 | 583.60 | 390.19 | 874.67 |
| **Townson** | | | | |
| 10 | 434.12 | 419.51 | 289.18 | 303.4 |
| 20 | 528.48 | 513.25 | 344.46 | 365.64 |
| 50 | 670.93 | 657.57 | 424.26 | 460.33 |
| 100 | 796.57 | 788.64 | 493.24 | 544 |
| **Whitestone** | | | | |
| 10 | 311.75 | 325.49 | 285.94 | 285.94 |
| 20 | 370.14 | 390.83 | 340.92 | 340.92 |
| 50 | 452.33 | 479.13 | 422.21 | 422.21 |
| 100 | 512.60 | 546.52 | 499.93 | 499.93 |

Table 4 shows the variation in gridded rainfall data for the 100-year return period from 450.77 mm (non-stationary) to 441.30 mm (stationary) for Gatton and from 575.84 mm (non-stationary) to 478.62 mm (stationary) for Helidon. For the design rainfall in Helidon (gridded data), for a 100-year period, the change between stationary and non-stationary was more noticeable; 97.22 mm compared to 9.45 mm in Gatton. Whereas in Helidon (station data), the difference between stationary and non-stationary indicates less variability, 17.4 mm (100-year period) compared to Gatton, 73.99 mm (100-year period). Of the 14 stations, the GEV distribution rainfall shows that 5 stations' data (Helidon, Gatton, Placid hills, Townson and Whitestone) and 3 gridded data (Helidon, Gatton and Placid hills) represent the non-stationary condition, respectively (Table 4). As Gatton is identified as the agricultural hub in Southeast Queensland, it is important to understand the short, intense rainfall for water resource management. As can be seen from Table 4, Gatton station represents the most remarkable variations in comparison to other stations. Rainfall return levels at high return periods were found to have noticeable variation compared to low return periods. The ERA5 estimates are noticeably lower than the station-based estimates for the region, as shown in Table 4, especially at the 50 and 100-year return periods.

### 3.5. Results of Stationary and Non-Stationary Analysis for Evapotranspiration

For the management of water resources, a precise estimation of evapotranspiration is necessary. Among the ET algorithms, SEBAL is the most promising methods for estimating evapotranspiration [18]. The GEV distribution evapotranspiration confirmed that, of the 14 stations, 6 stations' data (Helidon, Gatton, Placid hills, Thornton, Townson and Whitestone) and 2 gridded data (Thornton and Whitestone) exhibited the non-stationary condition.

Figures 6 and 7 show the return levels for evapotranspiration derived from the geeSEBAL algorithm for both station and global climate data. According to Figure 6a–e, stations' actual evapotranspiration shows a range of 184.88 mm (stationary) to 169.02 mm (non-stationary) for the 100-year return period and from 169.38 mm (stationary) to 157.86 mm (non-stationary) for the 10-year period in Gatton. Helidon also exhibits variation in $ET_a$, ranging from 189.76 mm (stationary) to 161.98 mm (non-stationary), from 164.43 mm (stationary) to 153.41 mm (non-stationary) for (100-year period) and (10-year period), respectively. The difference between stationary and non-stationary reaches 27.78 mm in Helidon compared to 13.99 mm in Townson for the 100-year return period. Additionally, this demonstrates that Helidon experiences significant variability compared to other stations. According to Figure 7a-d, $ET_a$ derived from the geeSEBAL algorithm ranges from 171.32 mm (stationary) to 170.62 mm (non-stationary) for the 100-year period and from 149.57 mm (stationary) to 146.22 mm (non-stationary) for the 10-year period in Thornton. The least variation is observed from the 100-year to 10-year return level periods.

### 3.6. Results of Stationary and Non-Stationary Analysis for Water Storage

Due to a lack of field data for water storage, this variable was assessed based on the satellite-derived images. The return levels of water storage for station and global climate data are shown in Table 5. The 10-year and 100-year return levels' water storage derived using the global climate data varies from 202.44 mm to 224.80 mm and from 95.45 mm to 108.70 mm for stationary and non-stationary, respectively in Gatton. The results of the Gatton station data illustrate no variation between the stationary and non-stationary conditions. Whitestone and Gatton act quite similarly when experiencing non-stationary conditions in the global climate data. The GEV distribution for water storage in Whitestone station varies from 321.33 mm (stationary) to 344.60 mm (non-stationary), from 99.57 mm (stationary) to 106.32 mm (non-stationary) for (100-year period) and (10-year period), respectively, whereas the rest of the stations indicate a lack of variation in the stationary condition.

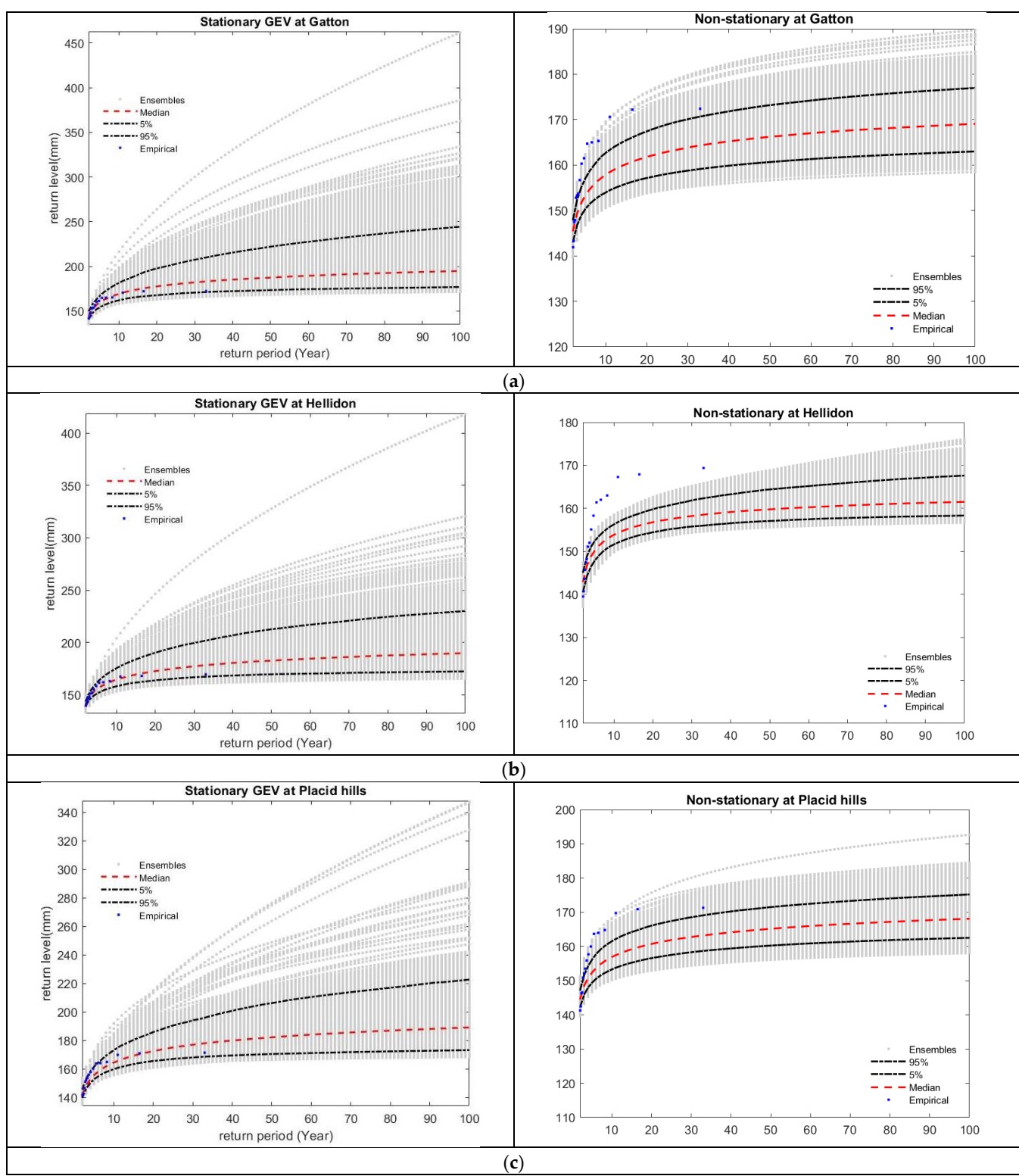

**Figure 6.** *Cont.*

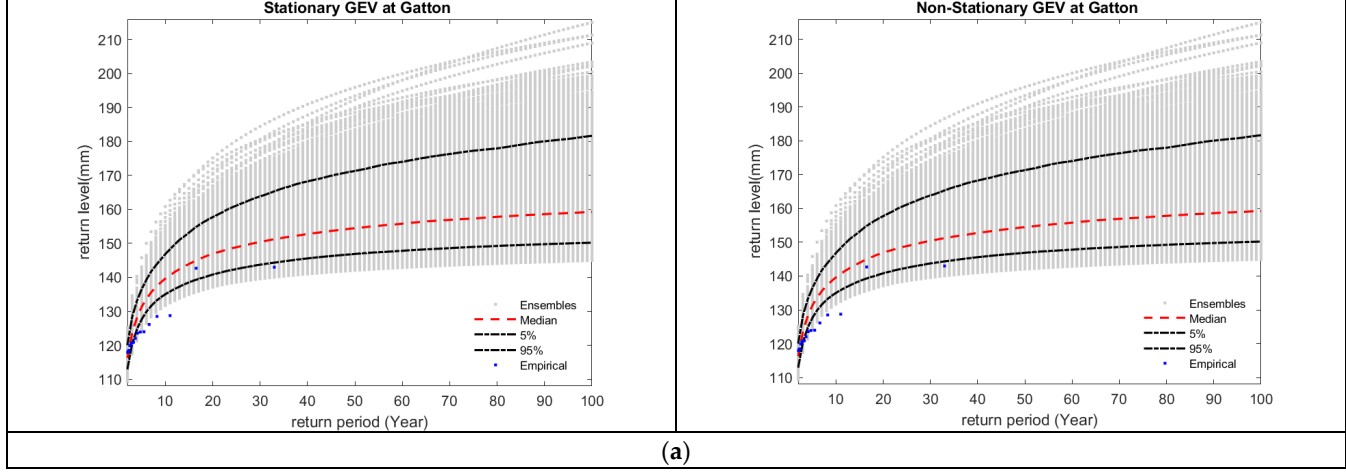

**Figure 6.** The output of NEVA's non-stationary GEV framework, standard return levels with design exceedance probability for $ET_a$ based on station data (**a–e**) (Figure generated using MATLAB).

**Figure 7.** *Cont.*

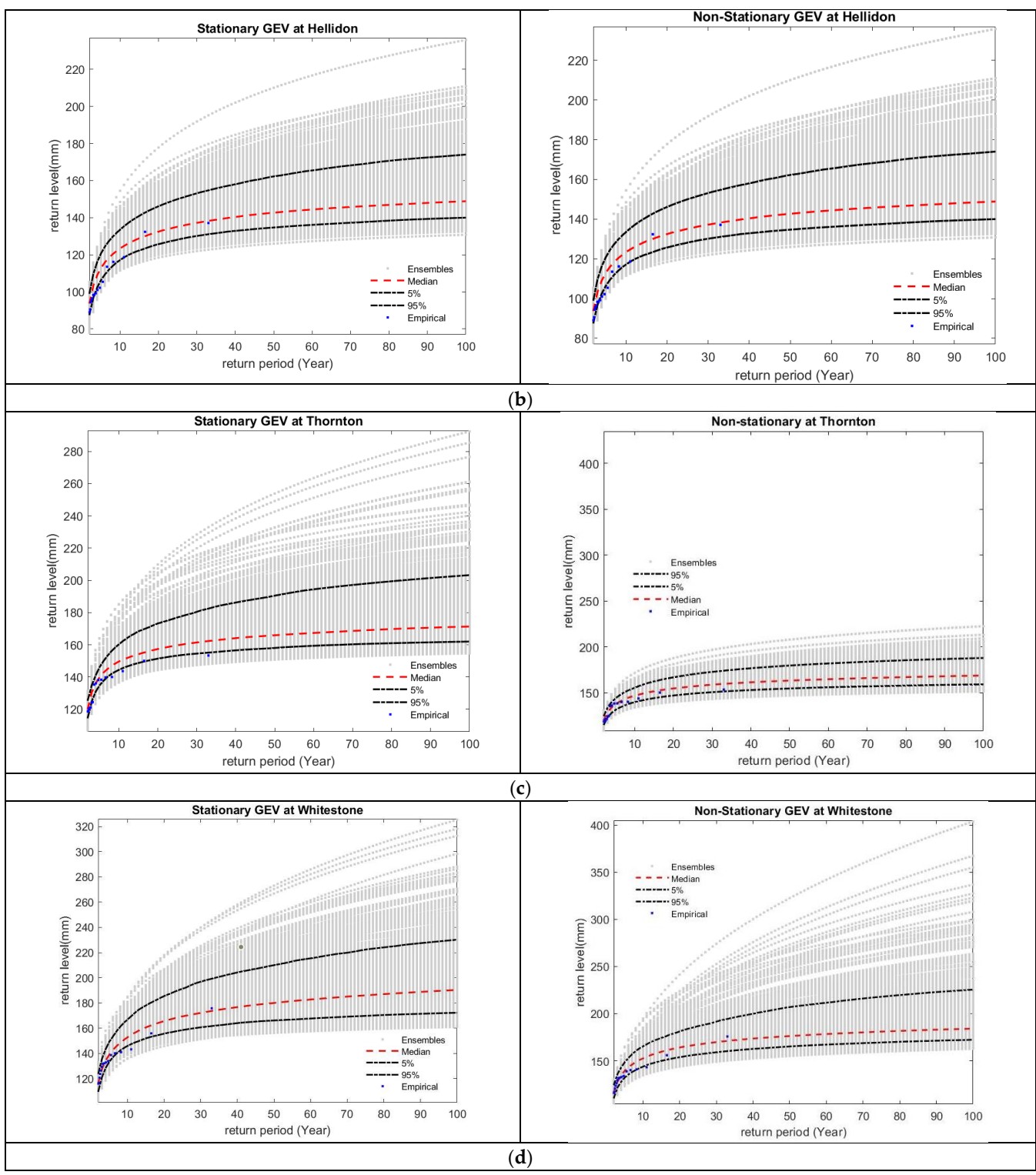

**Figure 7.** The output of NEVA's non-stationary GEV framework, standard return levels with design exceedance probability for $ET_a$ based on global climate data (**a–d**) (Figure generated using MATLAB).

**Table 5.** The maximum water storage in different return periods of different stations in the Lockyer Catchment.

| Return Period | Water Storage Based on Station | | Water Storage Based on ERA5 Data | |
|---|---|---|---|---|
| **Gatton** | **Stationary** | **Non-Stationary** | **Stationary** | **Non-Stationary** |
| 10 | 100.71 | 100.71 | 95.45 | 108.70 |
| 20 | 144.89 | 144.89 | 126.97 | 142.55 |
| 50 | 202.65 | 202.65 | 168.84 | 188.3 |
| 100 | 247.64 | 247.64 | 202.44 | 224.80 |
| **Helidon** | | | | |
| 10 | 133.93 | 133.93 | 132.96 | 132.96 |
| 20 | 196.26 | 196.26 | 156.91 | 156.91 |
| 50 | 286.70 | 286.70 | 195.64 | 195.64 |
| 100 | 363.34 | 363.34 | 224.83 | 224.83 |
| **Placid Hills** | | | | |
| 10 | 120.79 | 120.79 | 113.70 | 113.70 |
| 20 | 179.07 | 179.07 | 136.92 | 136.92 |
| 50 | 255.67 | 255.67 | 163.25 | 158.23 |
| 100 | 316.06 | 316.06 | 180.55 | 176.99 |
| **Thornton** | | | | |
| 10 | 165.76 | 165.76 | 110.55 | 110.55 |
| 20 | 233.13 | 233.13 | 137.04 | 137.04 |
| 50 | 328.51 | 328.51 | 174.03 | 174.03 |
| 100 | 415.159 | 415.159 | 199.66 | 199.66 |
| **Townson** | | | | |
| 10 | 197.20 | 197.20 | 97.80 | 97.80 |
| 20 | 256.35 | 256.35 | 136.77 | 136.77 |
| 50 | 343.19 | 343.19 | 193.38 | 193.38 |
| 100 | 411.08 | 411.08 | 241.17 | 241.17 |
| **Whitestone** | | | | |
| 10 | 99.57 | 106.32 | 96.92 | 96.92 |
| 20 | 156.79 | 164.97 | 147.03 | 147.03 |
| 50 | 244.30 | 258.30 | 231.33 | 231.33 |
| 100 | 321.33 | 344.60 | 314.13 | 314.13 |

## 4. Discussion

This research study's results clearly indicate that extreme events including extreme rainfall, extreme evapotranspiration and water storage deficit under the stationary assumption are mostly lower than the extreme events under the non-stationary assumption in both station and global climate datasets. Like previous studies on rainfall IDF [60,61], the annual rainfall extremes, evapotranspiration extremes and water storage extremes are well-described by the GEV distribution. In a previous study, the global-scale models of climate variability were considered in the development of nonstationary TDF curves in Canada [28]. Another study on rainfall IDF shows that adoption of stationary assumption underestimates the design climate extremes [23]. Since rainfall extremes are well-researched

from both the stationary and nonstationary perspectives [62,63], a comparison can easily be made. This study considered stationary and non-stationary assumptions. The reasons for adopting this approach are its easy estimation procedure which is suitable for professional applications. In the majority of instances, the gap between stationary and non-stationary widens as the return period increases as it is highlighted in several studies [1,64]. Numerous studies have been performed around the world to prove the effect of non-stationary on extreme values [65,66]. Moreover, the extreme events in all studied variables and return periods of the global climate data are lower than the extreme events of the ground-based measurements. The spatial patterns derived from ERA5 land reanalysis data closely resemble the station data for water storage and rainfall. In $ET_a$, ERA5 reanalysis and station were approximately the same. However, the ERA5 land data of some years overestimates the higher spatial variability compared to station data. Therefore, reanalysis data offers satisfactory outcomes for estimating $ET_a$, providing an intriguing option for assessing spatial and temporal aspects across regional and continental scales.

A common pattern of uncertainty ranging between stationary and non-stationary assumptions is noticed across all stations with varying variables, including extreme rainfall, extreme $ET_a$ and extreme water storage deficit. The findings suggest that extreme rainfall shows a greater disparity between ground-based measurements made under stationary and non-stationary assumptions compared to data obtained from global climate products. The uncertainty range expands from the 10-year period to 100-year period, with the water storage deficit showing less variation between stationary and non-stationary conditions, while rainfall exhibits a sharp variation, and evapotranspiration shows moderate variation. Evapotranspiration and water storage derived from ERA5 land reanalysis data follow the same uncertainty with stations' data. The highest difference between stationary and non-stationary 100-year station rainfall is observed in Thornton (89.25 mm), Gatton (73.99 mm), Whitestone (33.92 mm), Placid Hills (23 mm) and Helidon (17.67 mm), respectively. It is noted that the global climate data tends to underestimate the outcomes in comparison to station data. Overall, this study highlights the importance of considering non-stationary conditions in hydroclimatic analysis and provides valuable insights into the performance of the geeSEBAL algorithm and NEVA methods for estimating and analysing hydrological variables.

## 5. Conclusions

In this study, the non-stationary and stationary conditions for extreme rainfall, extreme evapotranspiration and water storage deficit were investigated in Southeast Queensland (SEQ), Australia. The geeSEBAL algorithm was used to generate actual evapotranspiration, and the NEVA programme was used to estimate the intensity and frequency of extremes using the GEV distribution. The estimation was likely based on the assumption that the location parameter has a linear relationship with time. The key findings of this study are as follows: (1) The $ET_a$ derived from geeSEBAL by feeding global climate products overestimates $ET_a$ derived from geeSEBAL by feeding station data; (2) the statistical results indicate reasonable accuracy agreement between global climate datasets, $P$, $ET_o$ and geeSEBAL $ET_a$ and ground-based measurements; (3) the analysis of the spatiotemporal distribution of rainfall is almost identical with two different climate datasets (ground-based observations and global climate products). $ET_0$ from global climate datasets underestimates the results compared to station data. Interestingly, rainfall and $ET_a$ have shown that the highest rise is in the year 2010, which admits the historical flooding in SEQ; (4) the spatiotemporal distribution of water storage derived from a lump water balance remains the same for both climate datasets; (5) the intensity and frequency of rainfall extremes, evapotranspiration extremes and water storage deficit extremes under the stationary assumption are mostly lower than extreme events under the non-stationary assumption in both ground-based and global climate datasets under different return levels.

This research utilised the geeSEBAL algorithm in the Google Earth Engine (GEE) environment to estimate evapotranspiration ($ET$) at regional scales using Landsat imagery.

Understanding the dynamics of evapotranspiration (*ET*) is essential to address the challenges related to freshwater scarcity and increased water demand for agriculture and food production. While the GEV distribution approach was employed to analyse extremes, further investigations into other mechanisms, such as evapotranspiration and water storage, would provide a more comprehensive understanding of the underlying issues. The examination of station data also demonstrated that, even though geeSEBAL primarily relies on reanalysis data for meteorological input, the estimations of ET are similar to those obtained when the algorithm is supplied with real ground-based observations of meteorological conditions. The evaluation of $ET_a$ and water storage deficit demonstrated that the geeSEBAL algorithm has the capability to enhance irrigation management in Gatton, a prominent agricultural centre in the study area. By addressing the rising water needs for food production and water supplies, it also has the ability to lessen the consequences of drought [45]. This research can provide a foundation for future research and facilitate comparisons with existing studies by focusing on advanced statistical analysis. Understanding the statistical properties and characteristics of extreme events is essential for risk assessment and management and for the development of adaptation strategies. From the results, it can be concluded that extreme events under the stationary assumption are lower than extreme events under the non-stationary assumption in both global climate products and ground-based measurements for this region. In this study, the framework used examined the severity and coverage of various variables by analysing their frequency, intensity and duration, thereby exploring their spatiotemporal evolution. This framework can be applied to any geographical area, providing estimations of extreme conditions that are essential for infrastructure planning and design, risk assessment and disaster management. As we only covered the non-stationary conditions in extreme events with NEVA by incorporating the outputs of geeSEBAL, it is recommended that, in future studies, spatial Bayesian Hierarchical Modelling methods and ProNEVA [67] that allow for the incorporation of additional covariates for modelling the spatial variability observed in the GEV parameters are applied.

The outcomes regarding extreme events highlight that failure to account for non-stationary conditions and neglecting trends in extreme events results in inaccuracies when estimating such events. These inaccuracies can lead to errors in infrastructure development and design, causing financial losses and potential harm to human lives. Conducting detailed investigations into the underlying causes of extremes can be interesting for future studies. By quantifying the probabilities of extreme occurrences, decision-makers can make informed choices regarding infrastructure development, resource allocation and emergency planning.

**Author Contributions:** Conceptualization, Hadis Pakdel, Dev Raj Paudyal, Sreeni Chadalavada, Md Jahangir Alam and Majid Vazifedoust; Methodology, Hadis Pakdel, Dev Raj Paudyal, Sreeni Chadalavada, Md Jahangir Alam and Majid Vazifedoust; Software, Hadis Pakdel and Majid Vazifedoust; Validation, Hadis Pakdel; Formal analysis, Hadis Pakdel and Majid Vazifedoust; Investigation, Hadis Pakdel; Resources, Hadis Pakdel; Data curation, Hadis Pakdel; Writing—original draft preparation, Hadis Pakdel; Writing—review and editing, Dev Raj Paudyal, Sreeni Chadalavada, Md Jahangir Alam and Majid Vazifedoust; Visualization, Hadis Pakdel; Supervision, Dev Raj Paudyal, Sreeni Chadalavada and Md Jahangir Alam; Project administration, Dev Raj Paudyal, Sreeni Chadalavada and Md Jahangir Alam. All authors have read and agreed to the published version of the manuscript.

**Funding:** This research received no external funding. This research was supported by the Graduate Research School, University of Southern Queensland and this is part of the first author's PhD project entitled "Variability of Extreme Climate Events and Impacts of Future Climate Change on the Streamflow".

**Data Availability Statement:** The data supporting the findings of this study are available from the first author upon reasonable request.

**Acknowledgments:** The authors would like to thank the reviewers and the editors who gave valuable time to review the manuscript.

**Conflicts of Interest:** The authors declare no conflict of interest. Jahangir Alam works at the MDBA; however, this research has no link with the MDBA.

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
