# Peer review of "A Multi-Framework of Google Earth Engine and GEV for Spatial Analysis of Extremes in Non-Stationary Condition in Southeast Queensland, Australia"

_ijgi, doi:10.3390/ijgi12090370_

Round 1
Reviewer 1 Report
Hello,
The article is well organized and well written except for some inconsistencies in notations in symbols and description of figures which can be significantly improved with minor editing. The followings are some suggestions.
1. Line: 213: rah seems to have typos as it should have a suffix as in equation (2)
2. Line: 217, typo Ts based on the equation (s). The same appears in the flow chart Figure 2
3. Equations 5 and 6 should have errors.
4. Equation 9: formatting error.
5. Notation consistency is recommended such as ET0 / ETa throughout the text such as lines 373 and 412.
6. Consistency in representation in hypothesis in Lines 438 and 439.
7. Figure 6 return periods, it is unclear what the gray color represents. If “Ensembles” in the legend represent it, it is good to have it clearly visible. Legends and figures are overlapped in Figures 6 and 7 and non-overlapping is recommended.
Well organized and well written.
Author Response
Dear Reviewer
Dear Reviewer,
I am writing to inform you that the response draft has been prepared and is now attached as a Word document for your kind consideration. I grately appreciate your time and expertise in reviewing our submission.

Reviewer 2 Report
"google earth engine" should be written as "Google Earth Engine"
The first sentence of the abstract is not clear. Which extremes?
What is GEV? Write the long version in its first appearance.
The second sentence of the abstract is very long and hard to follow.
What is NEVA?Write the long version in its first appearance.
The abstract is not well written. The authors should revise it. First, they need to better explain the problem. Then the importance of the selected region and used dataset should be mentioned. Afterwards, the methodology and the novelty should be addressed. Finally, the most important quantitative findings should be emphasized.
The introduction of the paper is not comprehensive and coherent, it needs to be rewritten. There is not a good logical design between the transitions of the consequential paragraphs. The last paragraph of the introduction should clearly explain the objectives of the study and the literature gap that this paper is filling. The novelty of the paper is not clear.
Authors should clearly explain why "Lockyer catchment" is selected and why t should receive global interest.
The authors should mention the number of meteorological stations.
Figure 1 is not clear. Figure 1(a) should include a better map and some annotations. The markers for the stations should be bigger in Figure 1(b).
Figure 2 should be revised. The logic is not clear.
For sub-figures, authors should use (a), (b), (c) etc.
Figure 7 covers 4 pages. It needs to be divided.
The discussion section is very weak. Authors should comperatively discuss their findings with other researchers.
The conclusion simply repeats the results. They need to revise it. There should be a scientific message here. Some future insights, drawbacks of the proposed methodology shuld be provided.
Moderate editing of English language required
Author Response
Dear Reviewer,
I am writing to inform you that the response draft has been prepared and is now attached as a Word document for your kind consideration. We greatly appreciate your time and expertise in reviewing our submission.

Reviewer 3 Report
Reviewer Comments
Title: Which extreme event type are you modeling? This should be clear from the start.
Abstract: Notes that they are analyzing catchment conditions and extreme events… however, there is no mention within the abstract of what they mean by “extreme events” (Line 18) – do the authors mean extreme flooding events, extreme precipitation events, or some other extreme? This needs to be clearly defined for the reader.
e.g., Line 23 states “spatial and temporal variation within Lockyer Valley” – but spatiotemporal variation of what? The variables that are being modeled should be stated earlier in the Abstract.
Also, are you using NEVA or the newer ProNEVA (process-informed NEVA) released in 2019?
Line 26: How is an increasing uncertainty range a novel concept? This is well known, since the rarer an event is (e.g., 100-yr event) the less information we have about it and the more uncertainty we have in our estimates. This was established long ago, cite appropriate sources.
Line 27-28: About the “less variability observed between stationary and non-stationary conditions in soil moisture”, this is also the result of the physics of the soil properties, no? The idea of extreme soil moisture is a big question here that needs to be addressed... since for rainfall and ET, you can have ever-increasing amounts so it is simple to select the annual maximum. But soil moisture has sharp cut-offs at 100% saturation for its maxima, and at 0% for its minima. So, how do you pick one annual maximum/minimum?? It's pretty limited in that way; however, extremes in the duration of extremely low or high soil moisture could be modeled. That would be similar to how we model drought (lack of rain) using extreme value theory. We have to either model the duration or intensity of the drought since zero rainfall isn't necessarily extreme on its own.
Line 28: “moderate variation in evapotranspiration” – yes, ET is dependent on and limited by temperature, vegetation, and holding capacity of the atmosphere (e.g., relative humidity). Have you considered using spatial Bayesian Hierarchical Modeling methods that allow for the incorporation of additional covariates for modeling the spatial variability observed in the GEV parameters?
Line 30: “underestimating the amounts” of what? Directly state the variables that your results indicate this for.
Introduction: The introduction needs work to not confuse readers. It is a lot of information, but it is not always clear how this information relates directly to the study being conducted and the methods being used. The start of the introduction should introduce the “why” or motivation for the authors’ research, the past work that is similar and is being built upon, gaps in the past work that this study addresses (this is especially important) and conclude by highlighting the research that will be presented in the rest of the manuscript. Currently, there are too many other details to be able to clearly follow which extreme event type(s) will be analyzed and why, and which methods specifically (e.g., block maxima or peaks over threshold, stationary or nonstationary, single site or spatial methods) are being discussed.
Line 41-42: This is *not* a sufficient explanation for the definition of “extreme” for the purpose of GEV modeling methods. The authors did state that they will use the GEV distribution. The GEV does not use threshold exceedances, that would be the Generalized Pareto Distribution (GPD). The comments in this line are confusing and sound as if the authors are using threshold exceedances. GEV uses block maxima; therefore, in the case of GEV, extreme is the largest event in a set block of time (e.g., 1 year). The authors should cite the proper references for what is considered extreme in the case of the GEV distribution that they intend to use.
For the best information and references on the originators of the GEV family of distributions and what is considered extreme, please consult the following reference (which is frequently cited by the research team that developed NEVA and ProNEVA, and I believe is the authors' reference [60]):
Coles An Introduction to Statistical Modeling of Extreme Values; Statistics; Springer: London, 2001; ISBN 978-1-4471-3675-0.
Line 93: “Since extremes depend on…” Which extremes? Be specific. Drought (if so, which kind)? Heatwave? Extreme precipitation?
Line 94: “… the origins of these extremes…” Do you mean the climatic drivers? Or do you mean that you plan to do an attribution analysis? The whole sentence, lines 93-96, needs to be revised for accuracy. Identifying the drivers of extremes helps us quantify, predict, and project extremes – however, it doesn’t help us mitigate climate change. Climate change mitigation is a whole different topic. A study looking at the mitigation of impacts would have to include CO2 sources (attribution analysis) and future projections (e.g., CMIP6 scenarios) to quantify the reduction (or amplification) of the extreme event type and changes in the social and/or infrastructural impacts.
Lines 103-105: What are endmembers? Not everyone who reads this will know this term.
Line 120: Why only use NEVA? ProNEVA is the updated (as of 2019) and expanded package that allows for process-informed models. See the below reference:
Ragno E., AghaKouchak A., Cheng L., Sadegh, M., 2019, A Generalized Framework for Process-informed Nonstationary Extreme Value Analysis, Advances in Water Resources, 130, 270-282, doi: 10.1016/j.advwatres.2019.06.007
Line 130: Is this what the study is modeling: (1) extreme rainfall, (2) extreme ET, and (3) extreme soil moisture storage? Is this for daily data or subdaily? Are they being modeled individually or jointly? Also, which aspects of these extremes are being modeled? Duration, frequency, and/or intensity?
Line 145: This sentence should be placed elsewhere in the introduction re you are discussing GEV methods since this paragraph should only be about your proposed analysis/study. Start this paragraph with the sentence that starts on line 147, “In this study, we propose…”
When discussing extreme value analysis methods, it is always good to note that statistical models are specifically for scenarios where we do not have enough information on all the underlying physical drivers and their relationships. The statistical distributions are also a way of estimating the true distribution of the data without having all the data (e.g., if we only have 30 maximum/extreme events). Happily, there are ways to incorporate geographical and climatic variables which we know are likely to have a spatial relationship to the extreme data being modeled to help improve model predictive performance.
Line 148: “…return level of extremes.” Which extremes? Be specific.
Line 148-150: Remove the sentence starting with, “The new framework…” Since this is already stated at the end of the paragraph in more detail and is out of place here (interrupts the flow of ideas).
Materials and Methods:
The goal of this section is to detail the “what” and “how” of the analysis so that someone else could reproduce your results or methods if needed. Therefore, it requires updating to include more details.
Lines 184-188: Information to logically connect these two sentences/concepts is missing. Why do you need to develop models for nonstationary rainfall and temperature for the nonstationary relationship between runoff and rainfall?
Lines 190-193: (1) The reason for selecting these specific variables is still not clear to the reader since we still do not know what exactly is being modeled and why these variables are important for modeling whichever extreme is being modeled. State specifically what you are modeling and why these variables are important. (2) Also, it needs to be clear which variables are ground-based observations and which are remotely sensed satellite products.
Lines 195-196: (1) Was the station data checked for quality? How did you handle missing data, anomalous data, and/or duplicate sites? (2) What kind of stations were these? Rain gauges? Streamflow? Wind? Something else? Figure 1 right – what are “Silo_Stations”? (Note that this is a good place in the manuscript to refer to the figure again.)
Lines 199-200: What was the landuse information used for? Why include this data set?
Lines 200-201: Which DEM model is this? Cite a source.
Section 2.2 (Starting line 190) and Section 2.3 (Starting line 206): Both sections need a proper transition sentence/paragraph at their start to seamlessly guide the reader from the previous topic to the new one in a logical way.
Line 213: (rah) should match the format in Eqn. 2, where the ah is subtext.
Section 2.3: It should be made clear at the start of the section that this is just an overview of the geeSEBAL methods that lead to the outputs (restate which variables are from geeSEBAL) the authors are using. Versus a method the authors are using to generate outputs of their own (or vice versa depending on how this was used). For example, some papers that use PRISM datasets in the U.S. are using the output datasets provided by Oregon State University, while other papers use the PRISM method or algorithm to produce raster data of their own. Either way is acceptable, but their papers must make this differentiation clear for their readers.
Line 226-227: Is this an action that the authors did, or was this already integrated into GEE by the developers of geeSEBAL? This distinction is not clear.
Lines 303-305: Confusing sentence structure. Needs revision for clarity. Or better yet - just delete it if it is saying the same thing as the sentence in lines 308-310.
Line 307: What is the assumption that q = 0.2(P) based upon? The cited reference from FAO is on crop yields in response to water and associated ET. The cited page (and the whole reference from what I could find) did not reveal an equation that indicated that the runoff coefficient for this watershed’s land cover type is 0.2. This needs to be verified with a different reference that contains a table of runoff coefficients.
Lines 313-319: Since this is a part of the NEVA package and there is little detail given here, this does not need to be in its own separate section. Just add this information after the sentence in lines 353-354, before talking about the Bayesian methods.
Line 321: This sentence needs to cite a reference for this definition.
Lines 323-325: It would be more accurate to say: “The simplifying assumption of stationarity is used… “. Note that stationarity is an assumption that was historically made to simplify the already complex statistics required. It was often a safe assumption since our data records were shorter in the past; therefore, not much change occurred over that short period of time. Researchers of the past also did not have the same computational power that we now have, so they relied on simplifying assumptions to make the math manageable. (This would be very, very good to mention in the Introduction section of this paper to highlight why the authors are so interested in using non-stationary methods (this is one main gap they are addressing). Namely, we have longer data records and a changing climate which are making the continued assumption of stationarity potentially risky.)
Lines 325-326: The grammar for this sentence needs work.
Lines 326-328: This cannot be stated in this way. It currently sounds as if the authors are saying that because the cited paper from 2020 studied non-stationarity in Lockyer Catchment, that a single flood event in 2022 occurred because of non-stationarity. Which is not accurate. This needs to be reframed more clearly. It can accurately be stated that: the frequency of flood events like the one in 2022 may be misestimated under a stationarity assumption given that previous studies [45] have shown the assumption to no longer be valid.
Lines 333-334: Do you mean Extreme Value Theory (EVT)? Extreme Value Theory is what the GEV and GPD probability distributions are based upon. See reference [14] for confirmation of this.
Equations 5: Is missing the very important expression that goes within exp{ }. See Coles, 2001, p. 47-48 for the correct GEV formulation with both its versions; one conditioned on shape > 0 and the other on shape <= 0.
Lines 343-344: This sentence needs to end with “… , respectively.”
Line 345: The missing expression from Eqn. 5 appears to be placed here incorrectly before the location parameter variable. It needs to be removed and put back into Eqn. 5.
Lines 345-347: The variables are wrong given their earlier definition. Mu is location, epsilon is scale, and sigma is shape according to lines 342-344.
Line 357, Equations 8-10: Also needs to cite [14] since this is the formulation from NEVA being outlined as illustrated in their paper.
Lines 361-362: Indicate which posterior is corresponds to the stationary case and which to the non-stationary case.
Results:
Start the results section out with an introductory sentences/paragraph to remind readers what this study is modeling. This will help transition the reader from methods (all the statistics) back to the application of the methods so that they do not have to scroll back to remember.
The results figures need work. See my comments for each below. Having easier to interpret figures (plots) will make the results discussion easier to follow and understand. It is just too much right now to look back and forth and compare so many maps. They really need to be summarized in plots to communicate your data better.
Figures:
Subplots in all figures need to be labeled using lowercase letters (e.g., a, b, c, …). Figures 6-7 many of the letters are missing or do not align with any plots.
Figure 1: Clean up the figure legends to remove underscores between words. This can be done easily in the layer properties within ArcGIS (which is what this looks like it was created with).
Figure 2: Which parts of this flowchart are the authors’ model and which parts are already a part of geeSEBAL, GEE, and NEVA? Do the colors mean something?
Figures 3 and 4: Data visualization needs improvement. Timeseries of maps are very hard to interpret. Explore other ways of visually summarizing all these maps clearly. It is hard for the human eye to compare spatial changes across so many plots. For example, these could be summarized as box plots (or violin plots) of the pixel values so we can clearly see the spread of the data for each year and variable in an easier to digest way. Then the maps can be placed in supplemental information if readers wish to analyze them more closely. This would also remove the visual bias introduced by the arbitrarily assigned color values (color breaks) across the continuous data. To minimize visual bias in the color breaks, the authors can switch to a continuous colorramp for the maps (since they are continuous data), and a divergent colorramp for the variable that has both negative and positive values centered on zero.
*** Really important! *** Lines 441-443: (1) If Placid Hills and Townson are the sites with non-stationary time series, how big of a difference is there in the results at those sites using stationary NEVA versus non-stationary NEVA? Can you see in the results that there is misestimation of the empirical GEV distribution when using one assumption versus the other? If so, by how much? This should be the main highlight of the results section since the whole basis of this paper is that accounting for non-stationarity is important. So, do the results confirm your hypothesis for this study region? (2) Also… why show all the other sites if they all have stationary time series?
Lines 452-457: Are all these values in a Table somewhere? This is hard to read through. A table would be a better format for us to see the values in, then you can refer to the table when you discuss what all these numbers mean in broader terms. What are these results telling us? Why are they relevant to the discussion? What do they demonstrate?
Discussion:
All I’m seeing are reports of differences, but not a discussion of what the differences mean. Did the non-stationary NEVA results fit the empirical GEV of the stations with non-stationary time series better than the stationary NEVA model? This was the big gap that this study was interested in assessing. So, what do the results reveal? What do they mean? This is not addressed here, but it should be the main thing.
Lines 525-537 & Lines 544-548: This is exactly what should be stated in the Introduction section, so the readers know exactly what you are studying and why, and which methods and why.
Grammar, concision, and flow of logic need work.
Author Response

(The authors gave the same response as above.)

Round 2
Reviewer 2 Report
The abstract is very long, it should be shortened.
You should include results of different researchers and compare them with yours and comprehensively discuss them in the Discussion section.
Conclusion is summarizing the result. Authors should revise it. They need to emhasize the most important finding, give a scientific message and some future insights.
Minor editing of English language required
Reviewer 3 Report
Please see the attached document for my comments and recommendations. Thank you to the authors for putting so much effort and work into revising the manuscript and addressing the reviewer's concerns.
